# 🏛️ UAV-Flow Colosseo: A Real-World Benchmark for Flying-on-a-Word UAV Imitation Learning

**Xiangyu Wang**[1,2,*]**, Donglin Yang**[1,2,*]**, Yue Liao**[3,4,*]**, Wenhao Zheng**[1]**, Bin Dai**[2]**,**
**Wenjun Wu**[1,2]**, Hongsheng Li**[4]**, Si Liu**[1,†]

[1]Institute of Artificial Intelligence, Beihang University
[2]Hangzhou International Innovation Institute of Beihang University
[3]National University of Singapore      [4]MMLab, CUHK
{wangxiangyu0814,yangdonglin,liusi}@buaa.edu.cn,

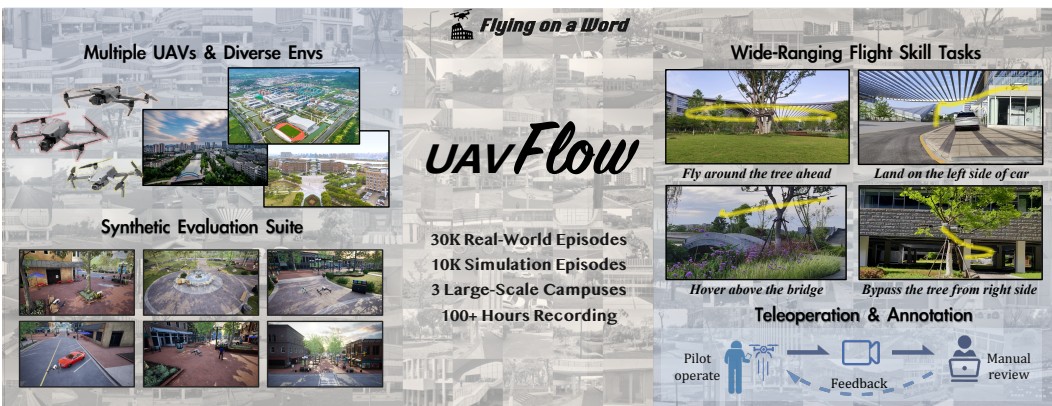

Figure 1: **Overview of our UAV-Flow benchmark.** It consists of a large-scale real-world dataset for language-conditioned UAV imitation learning, featuring multiple UAV platforms, diverse environments, and a wide range of fine-grained flight skill tasks. To enable systematic experimental analysis under the Flow task setting, we additionally provide a simulation-based evaluation protocol and deploy VLA models on real UAVs. To the best of our knowledge, this is the first real-world deployment of VLA models for language-guided UAV control in open environments.

## Abstract

Unmanned Aerial Vehicles (UAVs) are evolving into language-interactive platforms, enabling more intuitive forms of human-drone interaction. While prior works have primarily focused on high-level planning and long-horizon navigation, we shift attention to language-guided fine-grained trajectory control, where UAVs execute short-range, reactive flight behaviors in response to language instructions. We formalize this problem as the Flying-on-a-Word (Flow) task and introduce UAV imitation learning as an effective approach. In this framework, UAVs learn fine-grained control policies by mimicking expert pilot trajectories paired with atomic language instructions. To support this paradigm, we present UAV-Flow, the first real-world benchmark for language-conditioned, fine-grained UAV control. It includes a task formulation, a large-scale dataset collected in diverse environments, a deployable control framework, and a simulation suite for systematic evaluation. Our design enables UAVs to closely imitate the precise, expert-level flight trajectories of

---

[*]Equal contribution.
[†]Corresponding author.

39th Conference on Neural Information Processing Systems (NeurIPS 2025) Track on Datasets and Benchmarks.

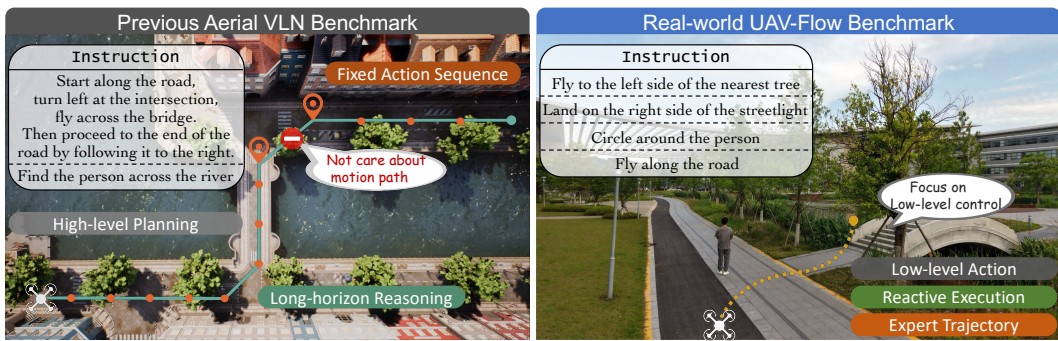

Figure 2: **Analysis of traditional UAV VLN and our Flow.** *Left:* VLN tasks aim to reach distant goals by planning long-horizon paths from instructions. *Right:* Flow focuses on executing short-range, language-guided trajectories toward visually grounded targets within the current scene.

human pilots and supports direct deployment without sim-to-real gap. We conduct extensive experiments on UAV-Flow, benchmarking VLN and VLA paradigms. Results show that VLA models are superior to VLN baselines and highlight the critical role of spatial grounding in the fine-grained Flow setting. As far as we are aware, we present the first real-world deployment of a VLA system for language-conditioned UAV control in open environments. Data, code, and real-world flight demos are available on `https://prince687028.github.io/UAV-Flow`.

# 1 Introduction

Unmanned Aerial Vehicles (UAVs), as the most popular low-altitude flying platforms, offer new perspectives for visual perception and assist humans in a wide range of tasks. With automated control algorithms [1, 2, 3, 4], UAV operation has evolved from a skill requiring expertise to an accessible, beginner-friendly technology. Today, a user can purchase a drone in the morning and capture cinematic footage by the afternoon. Beyond automation, the rise of large-scale AI models invites a new question: *"Can UAV manipulation become even more intuitive through language interaction?"* Imagine simply saying, "fly around me," and the UAV understands and acts accordingly. This shift from automation to intelligence marks a new frontier for human-drone interaction.

To enable language-interactive UAV control, recent research [5, 6, 7, 8, 9] has adapted vision-language navigation (VLN) tasks from ground robots [10, 11, 12, 13, 14] to aerial platforms, typically using simulated environments [15, 16] where UAVs interpret language instructions to search for targets or reach distant destinations, as illustrated in Fig. 2. These efforts primarily focus on high-level reasoning capabilities [17, 18, 19, 20] such as path planning and goal-directed navigation. They assume that low-level control, which executes short atomic flight behaviors such as moving between waypoints or responding to simple instructions, is already reliable for AI models. This assumption often holds for ground robots, but not for UAVs, which face the complexities of 3D flight, including high degrees of freedom and dynamic perspectives. Thus, language-guided low-level control emerges as a critical yet underexplored direction toward enabling intelligent UAV systems.

To operationalize Flow, we formulate *UAV imitation learning*, where the UAV learns to execute atomic language instructions by mimicking human pilot trajectories in real-world environments. As illustrated in Fig. 1, we introduce *UAV-Flow*, a benchmark for imitation learning of UAV control conditioned on language instructions, built around the Flow paradigm. It comprises a formal task definition, the first real-world dataset for language-conditioned UAV imitation learning, a real-UAV-deployable control framework, and a simulation suite for systematic evaluation.

To enable effective UAV imitation learning, we formulate the Flow task as mapping atomic language instructions to executable UAV actions, grounded in two core capabilities: *motion intent understanding*, which interprets low-level flight semantics (*e.g.*, "move 5 meters at a 45-degree angle"), and *spatial context grounding*, which links spatial references in language to visual observations (*e.g.*, "fly

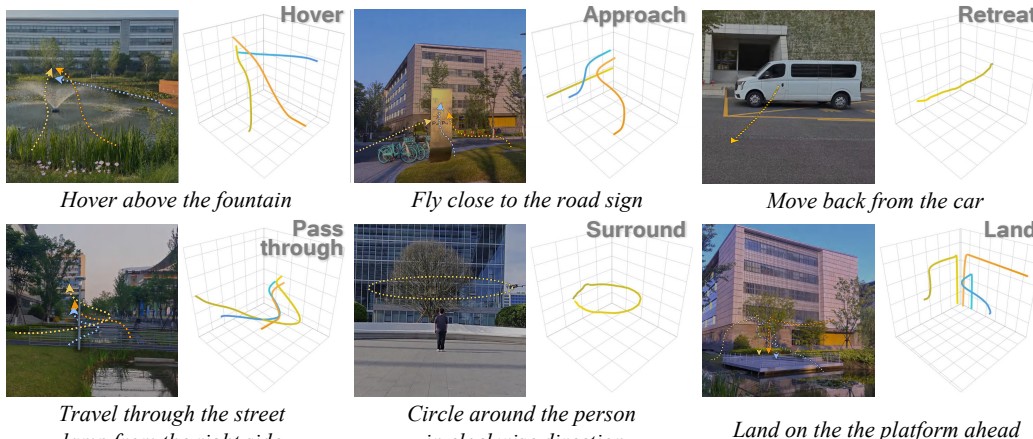

*Hover above the fountain*     *Fly close to the road sign*     *Move back from the car*

*Travel through the street lamp from the right side*     *Circle around the person in clockwise direction*     *Land on the the platform ahead*

Figure 3: **Visualization of Flow tasks.** Given the same instruction, human pilots execute diverse real-world trajectories. We show 2D flight paths over aerial scenes and reconstructed 3D trajectories.

to the right side of the marker"). Based on these capabilities, we define two corresponding task types that separately evaluate motion-level and perception-grounded execution.

To accurately capture the flexible and diverse flight behaviors exhibited by expert pilots, we depart from conventional simulator-based data collection paradigms and construct the UAV-Flow dataset directly in real-world environments. To the best of our knowledge, this is the first real-world dataset explicitly designed for language-conditioned UAV imitation learning. Moreover, it enables accurate imitation and direct deployment without a sim-to-real gap. Data is collected by professional pilots across three large-scale campus environments selected for their architectural diversity and spatial complexity. During collection, pilots perform flights by following language instructions within the visual context. We record synchronized UAV onboard video and corresponding 6-DoF state trajectories, resulting in comprehensive language-vision-action sequences.

While UAV-Flow provides a real-world foundation, deploying large-scale models onboard remains challenging. To address this, we propose a ground-drone collaborative framework, where the UAV streams state and visual inputs to a ground station for inference and receives control feedback with low latency impact. Additionally, to support systematic evaluation and controlled comparisons, we construct a simulation-based dataset under the same Flow task formulation.

We establish a comprehensive baseline benchmark for UAV-Flow by adapting representative methods from two paradigms: traditional VLN approaches [7, 10, 21] for high-level planning, and recent VLA methods [22, 23] designed for reactive control. These baselines are systematically transferred and evaluated under the Flow setting. Our experiments span both simulation and real-world deployments, enabling controlled comparisons and practical assessments. Results show that VLA models consistently outperform VLN models in fine-grained control, achieving more stable, deployable behaviors that align with the demands of real-world UAV systems.

## 2 UAV-Flow Benchmark

We shift the research focus for language-interactive UAV control from traditional *"flying far"* paradigms, which are centered on long-horizon path planning, toward *"flying better"*, which emphasizes short-range, fine-grained trajectory control. To realize this, we integrate an imitation learning framework into UAV control, enabling more precise and refined flight behaviors by mimicking the patterns of expert pilots. To support this paradigm, we introduce a large-scale, real-world benchmark for language-conditioned UAV imitation learning. We formalize the task setting and introduce a real-world dataset collected by professional UAV pilots, complemented by a simulation dataset for systematic evaluation under the Flow paradigm.

## 2.1 Flow Task Definition

To explore the *"flying better"* problem, we formalize a task that aligns language instructions with fine-grained, short-range flight execution, emphasizing visually grounded interactions and simple, human-like maneuvers (*e.g.*, orbiting or passing around obstacles). We introduce the *Flying-on-a-Word* (Flow) task setting, which evaluates a UAV agent's ability to translate language instructions into precise and dynamically feasible flight actions. Each Flow task instance provides the UAV with three modalities at every time step: a natural language instruction $\mathbf{I}$, the UAV's 6-DoF state $\mathbf{S}_t$, and an egocentric visual observation $\mathbf{O}_t$. The agent is expected to generate UAV action sequence that reflects the intent of the instruction while satisfying dynamic feasibility, thereby emulating maneuvers characteristic of expert pilots. To this end, we define the policy function:

$$\pi_\theta : (\mathbf{S}_t, \mathbf{O}_t, \mathbf{I}) \mapsto a_t, \tag{1}$$

where $a_t$ denotes the low-level control action executed at time $t$. Over the full execution of the instruction, the agent produces a sequence of control actions: $A = \{a_1, a_2, \ldots, a_T\}$, which collectively constitute the agent's response to the given instruction $\mathbf{I}$.

We identify two core capabilities essential for completing the Flow task: motion intent understanding and spatial context grounding. The former refers to the UAV's ability to interpret and execute basic flight behaviors, while the latter involves integrating visual perception with scene semantics to produce environment-aware trajectories.

Following this formulation, we define two instruction types: primitive motion commands and object-interactive commands. Primitive commands (*e.g.*, takeoff, translation, rotation, diving) evaluate the agent's ability to follow basic motion directives. Object-interactive commands (*e.g.*, approaching, orbiting, passing through, hovering) assess its capacity for perception-driven spatial reasoning. We present illustrative examples of trajectories corresponding to representative instructions in Fig. 3.

## 2.2 Real-World Data Collection

This section outlines the data collection and annotation pipeline used to construct the UAV-Flow dataset, as summarized in Fig. 4.

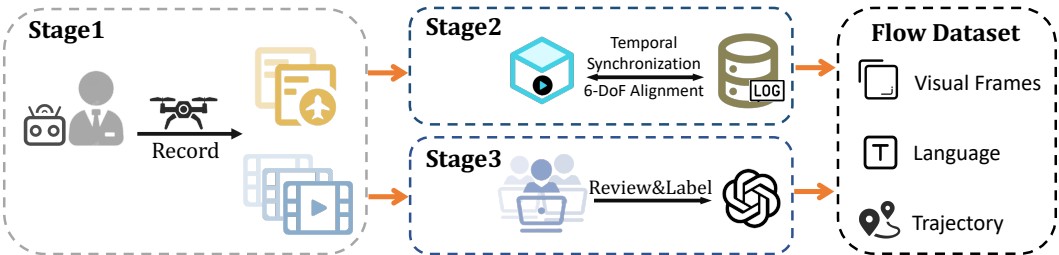

Figure 4: **Real-world UAV data collection pipeline.**

**High-Quality Trajectory Collection.** We collect a real-world language-conditioned UAV control dataset to support the Flow task, focusing on precise execution, perceptual alignment, and behavioral diversity. This data forms the foundation for subsequent instruction annotation and model training.

We conduct data acquisition across three university campuses spanning $5.02\,\mathrm{km}^2$. Each campus serves as a compact urban environment containing varied semantic elements such as pedestrians, vehicles, vegetation, buildings, and other landmarks, enabling rich visual contexts for diverse flight behaviors. All flights are manually operated by certified UAV pilots, each with over 800 hours of experience. We employ three commercial-grade DJI camera drones, Mavic 3T, Air 3S, and Mini 4 Pro, equipped with 4K cameras and RTK GPS modules, which provide high-resolution first-person-view video and centimeter-level trajectory accuracy, both of which are essential for trajectory reconstruction and multimodal alignment.

To guide flight execution, we curate a set of instruction templates spanning two categories: (1) primitive motion commands (*e.g.*, takeoff, shift, rotate), and (2) object-interactive behaviors (*e.g.*, "orbit around a car", "hover beside a landmark"). Pilots operate exclusively from the UAV's first-person view to ensure consistency between recorded inputs and model perception.

To enhance behavioral diversity, each instruction is executed from multiple starting positions. For example, the command "pass the car from the left" may begin from different relative angles, requiring the pilot to adapt trajectories while preserving semantic intent. This strategy-level variation enriches the dataset with functionally equivalent but visually diverse executions.

All flights are synchronously recorded with onboard video and complete flight logs, producing high-fidelity trajectories paired with visual observations for downstream annotation and learning.

**Trajectory-Visual Alignment.** Aiming for precise matching between images and flight paths, we synchronize raw flight logs with their corresponding aerial videos to construct frame-level pairs. Using timestamps, we align each 6-DoF state, including GPS coordinates (latitude, longitude, altitude) and orientation (roll, pitch, yaw), with the associated video frame via linear interpolation. For ease of fine-grained trajectory control, we transform global GPS coordinates into a local Cartesian coordinate system centered at the trajectory's starting position. Relative orientation is also computed with respect to the initial frame to capture heading dynamics. We uniformly sample the video at 5 Hz and pair each frame with its aligned UAV state, resulting in a high-quality sequence of visual observations and corresponding flight poses suitable for downstream learning and annotation.

**Language Instruction Annotation.** We organize a large-scale annotation team to review and label the flight videos. Annotators first filter out video segments with ambiguous or incoherent flight behavior. For the remaining valid clips, they compose precise and concise language instructions that describe the UAV's movement and its relation to the scene context. To enrich the diversity of language instructions and support both fixed-form and open-form instruction understanding tasks, we introduce a language diversification mechanism powered by large language models (LLMs). We first construct a Fixed Command Set with standardized descriptions for each task category, *e.g.*, all "side traversal" tasks are labeled as "fly through the right side of the object." We employ GPT series [24] models to enrich the base instructions, thereby creating an Open Vocabulary Command Set that includes diverse expressions.

Finally, we integrate language instructions, visual frames, and synchronized UAV flight states to construct our comprehensive language-vision-action multimodal dataset, UAV-Flow, designed to support fine-grained control tasks in real-world UAV scenarios.

## 2.3 Simulation Dataset under Flow Paradigm

To establish a unified evaluation benchmark, we follow the principles of Flow task and construct a simulation dataset named UAV-Flow-Sim within a UE-based campus environment. We utilize UnrealCV [16, 25] as the simulation environment for the UAV, controlling its motion through the control interface provided by the simulator. This control scheme closely mimics the position-mode control used in real-world UAV remote controllers, ensuring high fidelity to actual flight behavior. Moreover, UnrealCV supports a variety of placeable and movable interactive objects (e.g., humans, cars, quadruped robots), enabling the simulation of rich object interactions during data collection.

During the construction of the simulation dataset, we adopt a hybrid strategy. On one hand, human pilots manually collect flight trajectories by actively locating landmarks in the simulated environment. On the other hand, we leverage structured information available in simulation to implement rule-based data collection. Specifically, the UAV can perform distance-constrained maneuvers guided by its ground-truth state or using the target position within the scene to construct simulated data. We follow the same standardized pipeline as real-world data collection to construct the simulation dataset. Despite this, simulation environments still exhibit discrepancies from the real world in both visual perception and flight control dynamics. Therefore, we primarily use simulated data in virtual environments for model validation and analysis, while real-world UAV data is employed for training and deploying models in real-world scenarios.

## 2.4 Data Analysis

We construct two datasets based on the Flow paradigm: the UAV-Flow real-world dataset and the UAV-Flow-Sim simulation dataset. The real-world dataset contains 30692 flight trajectories categorized into 8 major motion types, each exhibiting diverse trajectory patterns. The distribution of motion types is shown in the left part of Fig. 5. We also provide a visual comparison between our dataset and prior VLN [5] dataset, highlighting the key distinctions—the long-horizon, discrete actions, and non-

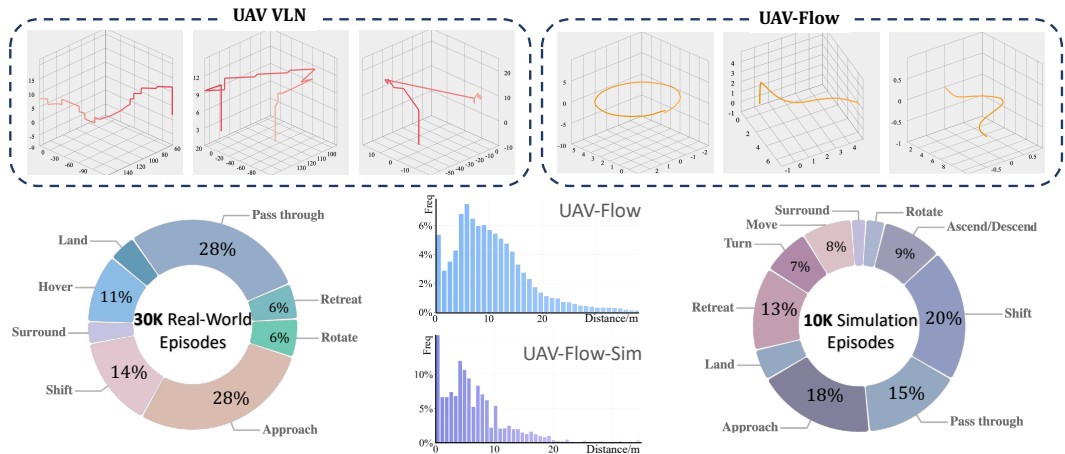

Figure 5: **Dataset statistics for UAV-Flow and UAV-Flow-Sim.** We show the distribution of task types (by percentage) and trajectory distances across both datasets.

dynamic simulation in traditional VLN tasks versus the short-range, fine-grained, and dynamically realistic execution in Flow task. The UAV-Flow-Sim dataset includes 10109 trajectories, built by referencing the typical actions in the real-world dataset and leveraging information accessible in simulation environments. The motion type is presented on the right of Fig. 5. Given that the Flow task emphasizes short-range, fine-grained control, most trajectories are within 20 meters in length, as illustrated in the center of Fig. 5. Notably, due to the inclusion of instructions such as in-place rotations, there remains a frequency of trajectories with near-zero displacement. Additionally, we construct a simulation test set including 273 annotated trajectories, covering all major action types in the simulation dataset, to facilitate systematic evaluation on the Flow task.

## 3 Flying-on-a-word (Flow) Colosseo

In this section, we introduce how to construct a *Colosso*—a unified "arena" for deploying and comparing UAV control algorithms in both real-world and simulated environments. To this end, we present a real-world UAV deployment strategy that enables large-scale model execution, along with a simulation-based evaluation suite designed for systematic comparison within the same task setting.

### 3.1 Real-World Ground-Drone Collaborative Deployment of Large-Scale Models

Limited onboard compute makes it infeasible to deploy large models directly on UAVs. Unlike stationary platforms, UAVs require lightweight, real-time control pipelines. As shown in Fig. 6, we adopt a ground-drone collaborative strategy, where the UAV streams FPV video and state data (via RTSP and MAVROS), and a ground station performs inference and returns low-level control actions over a wireless link. This setup introduces perception-action latency, which is particularly problematic for fast, continuous motion. Existing strategies include: (1) *Stop-and-Infer*, where the UAV pauses during inference but breaks task continuity; and (2) *Continuous Motion*, where the UAV continues moving but may suffer from delayed responses and control mismatch. To overcome these limitations, we propose a *Globally-Aligned Continuous Motion* scheme with a *look-ahead mechanism* for chunk-wise action prediction. Predicted target points are fused with the current UAV state to yield global poses. We further filter out already-passed targets based on UAV motion delay, improving control stability and execution accuracy under real-time constraints.

### 3.2 Closed-Loop Simulation Evaluation Metric

To evaluate the model's capabilities on Flow tasks, we develop a closed-loop simulation testing environment. We employ two metrics to evaluate the performance of baseline models: Success Rate (SR) and Normalized Dynamic Time Warping (NDTW) [26]. For each evaluation, we record the predicted trajectory and target point, render 2D and 3D visualizations, and determine the success rate

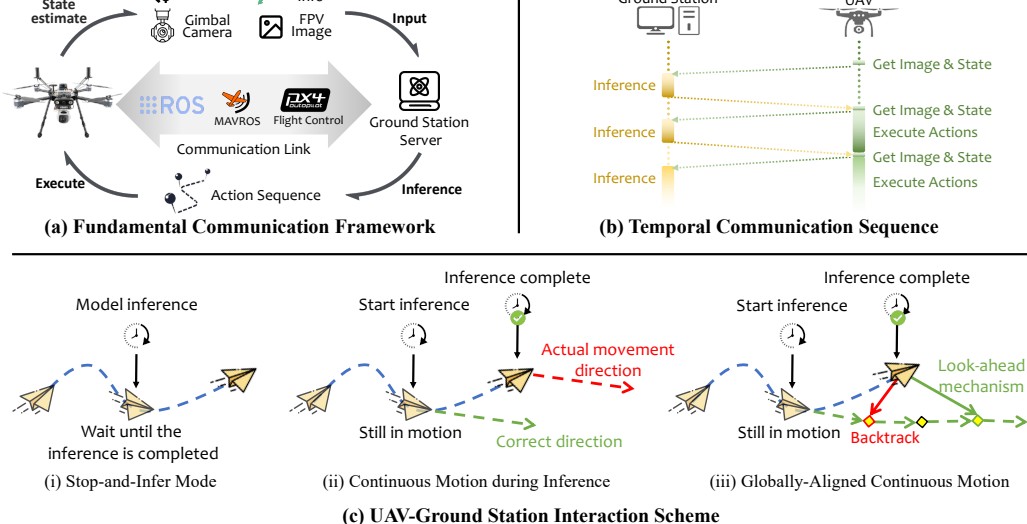

Figure 6: **Real-world UAV deployment of large-scale models.** UAV streams visual input and state to a ground station for inference, receiving control commands in return for real-time flight execution.

based on manual inspection of whether the trajectory semantically satisfies the instruction. Notably, some trajectories may be considered semantically correct yet follow suboptimal or irregular paths. To address this, we also compute NDTW to assess the similarity between the predicted and reference trajectories. In our implementation, we represent each trajectory point as a 6D vector by concatenating the position (x, y, z) with the cosine values of the orientation (roll, yaw, pitch), thereby capturing the influence of both position and orientation.

## 4 Experiments

We present a comprehensive experimental analysis of the UAV-Flow benchmarks. We adapt recent VLN and VLA methods to the Flow setting and evaluate them in the simulator. The selected methods are validated on the real-world UAV-Flow dataset and demonstrate real UAV deployments.

### 4.1 Benchmark Methods

We build our UAV-Flow benchmark upon representative model paradigms from recent VLN and VLA literature, and design task-specific adaptations to enable their effective use in UAV fine-grained control scenarios. Given the task discrepancy, where VLN models are designed for long-horizon navigation and VLA models for grounded robotic manipulation, we structurally modify and reconfigure these models to meet the unique demands of the Flow setting. The resulting models form a unified and extensible evaluation suite for benchmarking language-guided UAV imitation learning.

**VLN Models.** We first apply models originally designed for VLN tasks and adapt them to Flow tasks. Specifically, we adopt Seq2Seq [10] and CMA [21] as classical base models. Seq2Seq is a recurrent model that fuses image, instruction and previous action via a GRU to predict navigation actions. CMA uses a bidirectional LSTM to jointly encode image, instruction and previous action, and employs a cycle-attention mechanism to enhance performance. To adapt the models to our Flow task, we modify their original classification-based outputs over fixed discrete actions into continuous UAV pose regression. The resulting adapted versions are referred to as Seq2Seq-UAV and CMA-UAV. We also adopt the Travel [7] model, which is built upon the MLLM architecture for processing visual observations and textual inputs. By restructuring the input text to integrate both UAV state information and language instructions, and modifying the output to directly predict UAV poses from the fused feature, we obtain the adapted model termed Travel-UAV.

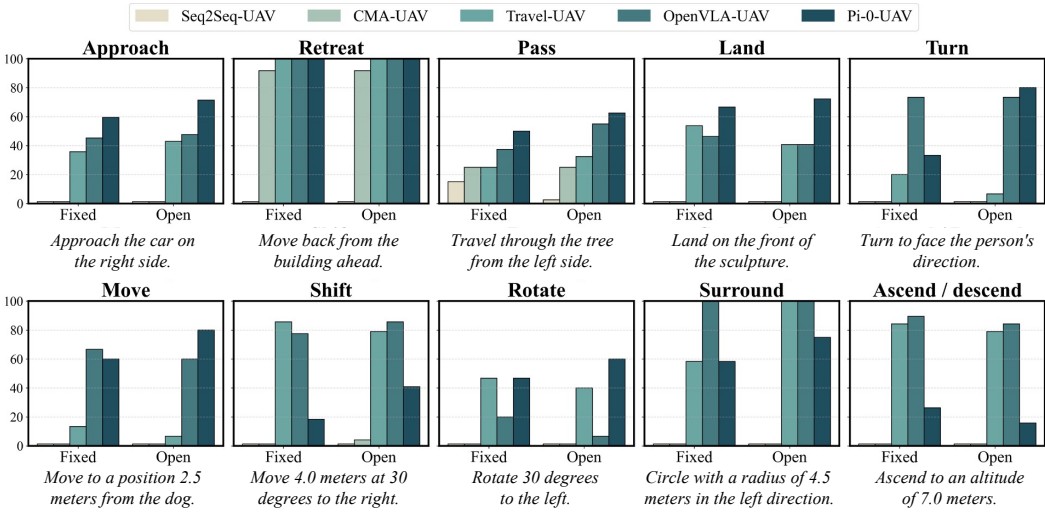

Figure 7: **Comprehensive evaluation on the UAV-Flow-Sim dataset.** We benchmark representative VLN methods and VLA methods from robotic manipulation across 10 Flow task types, reporting performance using the success rate (SR) metric.

**VLA Models.** We draw inspiration from recent advances in robotic manipulation and adopt Open-VLA [22] and Pi-0 [23] as base VLA models for Flow task. For OpenVLA, we retain its single-frame visual input design by feeding in the current image frame, while organizing the UAV state and instruction into a unified text input. The action space is discretized into 256 tokens, and the model predicts 6-DoF poses via token outputs. The adapted version is referred to as OpenVLA-UAV. For Pi-0, which supports multi-frame inputs, we use the first frame of each task as a reference and concatenate it with the current frame as visual input. The instruction and UAV state are encoded separately, and the model outputs a 6-DoF action chunk using a flow matching [27] mechanism. The adapted model is referred to as Pi-0-UAV. Further details of models are provided in Appendix B.

## 4.2 Results

We evaluate models on both the Fixed and Open Vocabulary Command Sets in a closed-loop simulation environment, as shown in Fig. 7 and Fig. 8. We observe that the VLN models, Seq2Seq-UAV and CMA-UAV, perform poorly on the Flow task. These models were originally designed for VLN over a discrete action space. When adapted to pose-level regression, they struggle to effectively fuse multimodal inputs for accurate pose prediction. Furthermore, due to their RNN-based architecture, the predicted trajectories tend to inherit the previous motion direction, often resulting in drifted or curved paths. These models also have difficulty predicting proper stopping points, causing the agent to continue moving indefinitely even after reaching the intended goal. These issues can be more clearly observed in trajectory visualizations in Appendix A.

Travel-UAV model differs from traditional VLN models by generating pose-level outputs directly from current-frame visual inputs, making it more suitable for the Flow task. It demonstrates strong motion intent understanding capabilities, effectively performing primitive motion instructions. However, built upon the LLaMA-VID [28] architecture, it encodes vision into only 17 tokens, limiting its ability to capture fine-grained visual semantics such as "turn to face the target".

OpenVLA-UAV demonstrates strong spatial understanding and motion execution on the Flow task. While it performs well in tasks requiring fine-grained motion control and visual grounding, its reliance on single-frame visual input imposes limitations in certain scenarios—for example, it may struggle to determine accurate stopping points when approaching specific sides of objects. Nevertheless, its overall visual perception remains strong, as shown in Appendix A, where it continues flying through two trees when failing to identify the termination condition.

Pi-0-UAV also demonstrates strong visual understanding and performs well in object-interactive tasks. However, constrained by the flow-matching training paradigm, it exhibits relatively weaker

performance on fine-grained motion intent instructions, as the flexible specification of distances and angles imposes higher demands on its semantic alignment capabilities—especially under limited training data. Additionally, its denoising-based inference mechanism may introduce slight fluctuations, compromising trajectory stability.

In summary, we observe that traditional VLN models are primarily designed around fixed and long-horizon action sequences, making them less suitable for the Flow task. Travel-UAV shares a similar paradigm with VLA models by generating waypoints from current visual inputs, but its limited capacity for fine-grained visual understanding hinders its performance in certain interactive tasks. In contrast, VLA models, originally developed for robotic manipulation tasks, demonstrate strong visual understanding and fine-grained control, leading to better performance in Flow task. Furthermore, we observe that training with the open vocabulary command

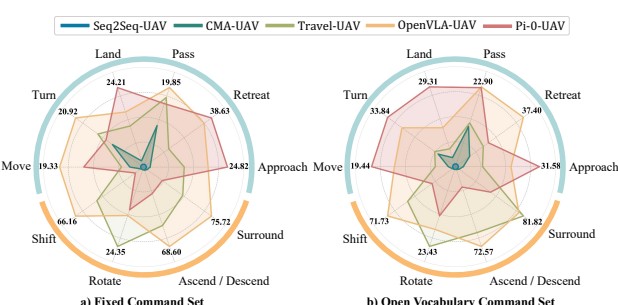

Figure 8: **Comprehensive evaluation on the UAV-Flow-Sim dataset based on the NDTW metric.** The outer light green area in the radar chart indicates object-interaction tasks, while the orange area represents primitive motion tasks.

set does not lead to an overall decrease in success rates for models like OpenVLA-UAV and Pi-0-UAV. Instead, it enhances their language generalization and even improves performance on certain tasks.

### 4.3 Real-World Deployment

Due to the safety constraints of UAV flight and the variability of outdoor environments, it remains challenging to adopt quantitative evaluation metrics in real-world scenarios. Nevertheless, we aim to verify the deployability of our model under real-world conditions. To this end, we draw upon the validation results from the simulation environment, and train the Pi-0-UAV model on the UAV-Flow real-world dataset. Deployment is conducted via our proposed ground-drone collaborative framework. The action chunk outputs of Pi-0-UAV integrate effectively with our look-ahead mechanism, enabling smooth and delay-free continuous flight control. In Fig. 9, we present representative flight examples and visualize the 3D trajectory sequences generated by the model, demonstrating its capability to execute actions accurately in real-world conditions.

## 5   Related Work

**Language-Guided UAV Tasks.**  Most existing language-guided UAV tasks fall under the VLN paradigm [5, 6, 7, 29], adapting navigation strategies from ground agents and focusing on high-level path planning. AerialVLN [5] provides sequential instructions for navigating long trajectories using discrete actions (e.g., move forward, ascend). CityNav [6] shifts toward goal-directed search but still relies on fixed action sets and long-range plans. Travel [7] introduces waypoint-level supervision

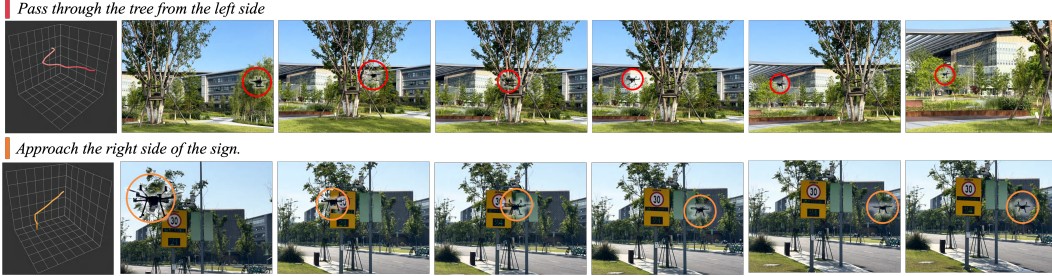

Figure 9: **Visualization of real-world UAV flight demos.** We deploy Pi-0-UAV, trained on UAV-Flow, on a real UAV and visualize the resulting flight trajectories alongside third-person dynamic views.

for more realistic control but still depends on external assistance for extended tasks. In contrast, language-guided low-level UAV control, which focuses on how UAVs react in real time to simple, fine-grained commands, remains underexplored. To address this gap, we propose the Flow task, which studies dynamic, low-level UAV behaviors grounded in natural language.

**Simulation vs. Real-World UAV Datasets.** Simulation environments [15, 16, 25, 30] are widely used in UAV research for their low cost, controllability, and ease of annotation. However, most UAV VLN datasets ignore flight dynamics, often collecting data from fixed drone positions. Some works simulate continuous motion using Unreal Engine with plugins like AirSim [15] or UnrealCV [16], but these setups still lack realistic flight dynamics and complex scenes. Meanwhile, real-world UAV datasets mainly focus on perception tasks [31, 32, 33, 34, 35], with limited work on language-driven dynamic control. To fill this gap, we collect a real-world language-vision-action dataset under the proposed Flow task and build a simulation system for controlled evaluation and benchmarking.

# 6 Conclusion

In this work, we introduce UAV-Flow, a novel benchmark designed to explore how imitation learning can enable UAVs to interpret language instructions and execute fine-grained dynamic motions. To support this effort, we collect a real-world dataset with 30k flight trajectories, covering a diverse range of motion types and environmental conditions. We further propose a ground-drone collaborative deployment framework that enables an end-to-end pipeline from data collection to model training and real-world UAV deployment. Additionally, we develop a complementary closed-loop simulation suite to facilitate systematic evaluation of model performance on the Flow task.

**Limitations.** Several limitations remain and warrant further exploration. On one hand, due to practical constraints such as flight safety and environmental variability, it remains challenging to conduct systematic real-world experiments and establish consistent evaluation metrics. In future work, we aim to develop a safer and fully closed-loop real-world evaluation framework to support comprehensive performance assessments. On the other hand, our current efforts primarily focus on short-range motion control within visual range. Integrating fine-grained short-range execution with long-horizon planning is a key challenge toward building truly intelligent, language-driven UAV systems, and will be a major direction for future research.

**Acknowledgements** This work was supported in part by the National Key R&D Program of China (No. 2022ZD0115502), the National Natural Science Foundation of China (No. 62461160308, U23B2010), the "Pioneer" and "Leading Goose" R&D Program of Zhejiang Province (No. 2024C01161), and the Ningbo Science and Technology Innovation 2025 Major Project (No. 2025Z034). It was also partially supported by the NSFC-RGC Project (No. N_CUHK498/24).

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

## A  Trajectory Visualization of Experimental Results

As shown in Fig. 10, We project the flight trajectories onto a 2D plane from a bird's-eye view and visualize examples for both object-interactive and primitive motion tasks. Overall, Seq2Seq-UAV and CMA-UAV struggle to interpret the motion semantics of the instructions, while Travel-UAV appears to learn fixed instruction-action mappings. In contrast, OpenVLA-UAV and Pi-0-UAV demonstrate stronger visual perception and motion capabilities, achieving more accurate instruction execution.

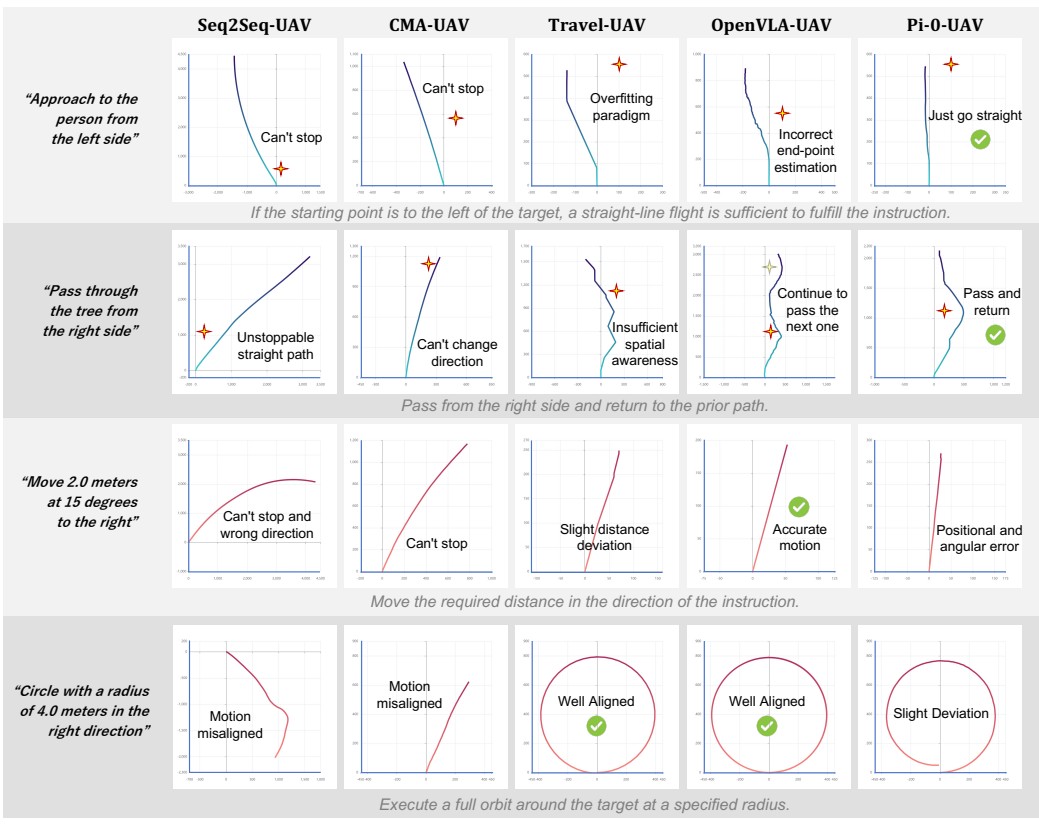

Figure 10: **Visualization of experimental trajectories.** We show representative examples from two categories: blue trajectories correspond to object-interactive tasks, while red trajectories illustrate primitive motion tasks.

## B  Model Structure

As shown in Fig. 11, We adapt and modify existing VLN and VLA models in detail to support the proposed Flow task.

For the Seq2Seq and CMA models, we replace their original discrete action classification outputs with continuous UAV pose regression. We also restructure the dataloader to match this formulation and adopt an MSE loss for training. While we experiment with incorporating the current UAV state (position and orientation of the UAV relative to the coordinate system of the first frame) as an additional encoded input concatenated with image and language features, we find that this leads to worse trajectory performance. As a result, we retain the original input design and construct the adapted versions named Seq2Seq-UAV and CMA-UAV.

For the Travel model, we preserve its visual encoder and modify the text input to integrate both UAV state and language instruction into a unified prompt. We also fine-tune the prompt template as shown in Fig. 12. On the output side, we modify the model to directly generate a sequence of 6-DoF UAV poses, resulting in Travel-UAV.

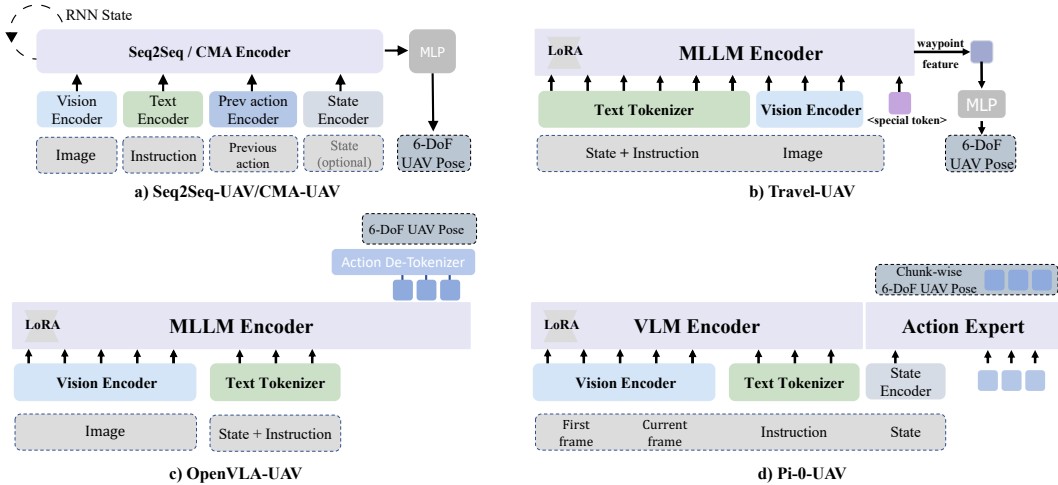

Figure 11: **Adapted model architectures.** We modify representative VLN and VLA models to support the requirements of Flow tasks.

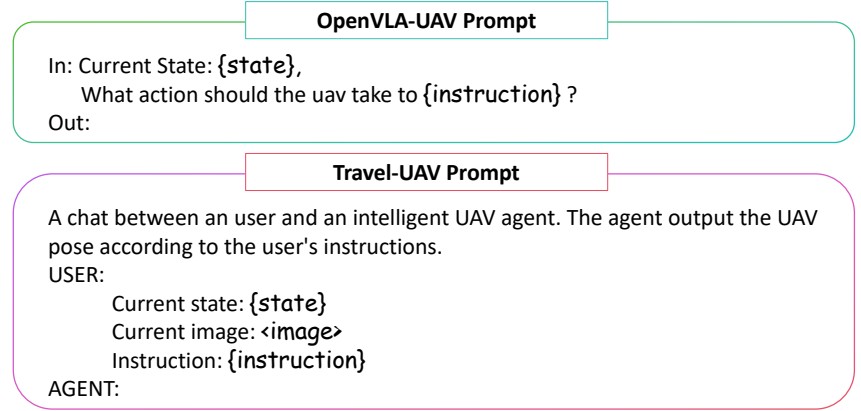

Figure 12: **Prompt templates for different models.** We illustrate how prompt formats are structured for OpenVLA-UAV and Travel-UAV.

For the OpenVLA model, we align the data format with its original setup used for robotic manipulation, and incorporate both state and instruction information as the textual input as illustrated in Fig. 12. This differs from the original design that used only instruction input. We maintain its single-frame visual input design and preserve the discrete 256-token-based prediction format for UAV poses prediction. This adapted version is referred to as OpenVLA-UAV.

For the Pi-0 model, we preserve its multi-frame visual input paradigm. Although the original Pi-0 uses multi-view images, our dataset contains only UAV FPV images. We address this by treating the first frame as a "reference frame" and combining it with the current frame for multi-frame input, suitable for short-range tasks. Additionally, we retain Pi-0's language and state encoders and train the model using the flow matching paradigm with a chunk size of 10. The resulting version is denoted as Pi-0-UAV.

## C   Training Implementation

We design a corresponding training strategy for each baseline model, minimising changes to the training settings in the original paper to ensure fairness and validity of the comparison.

| Parameter | Value |
|---|---|
| Batch Size | 32 |
| Epochs | 10 |
| Learning Rate | 1e-4 |
| GPU | $1 \times$ RTX 4090 |

Table 1: Seq2Seq-UAV and CMA-UAV training config.

| Parameter | Value |
|---|---|
| Epochs | 2 |
| Batch Size | 32 |
| Max Learning Rate | 5e-4 |
| LoRA | True |
| LoRA Rank | 32 |
| GPU | $8 \times$ A100 |

Table 2: Travel-UAV training config.

| Parameter | Value |
|---|---|
| Batch Size | 32 |
| Learning Rate | 5e-4 |
| LoRA | True |
| LoRA Rank | 32 |
| Max Training Steps | 200000 |
| GPU | $8 \times$ A100 |

Table 3: OpenVLA-UAV training config.

| Parameter | Value |
|---|---|
| Batch Size | 16 |
| Epochs | 12 |
| Learning Rate | 5e-5 |
| LoRA Enabled | True |
| LoRA Rank | 32 |
| Horizon Steps | 10 |
| GPU | $8 \times$ RTX 4090 |

Table 4: Pi-0-UAV training config.

## D  Inference Latency of Models

We measure the inference time of different models, covering the full pipeline from image input processing to trajectory generation. This provides a comprehensive reflection of the models' latency in real-world deployment. Notably, we set the Pi-0-UAV model to output an action chunk size of 10, allowing it to predict actions for 10 future time steps per inference.

| Model | Inference Latency (s) |
|---|---|
| Seq2Seq-UAV | 0.057 |
| CMA-UAV | 0.067 |
| Travel-UAV | 0.188 |
| OpenVLA-UAV | 0.172 |
| Pi-0-UAV | 0.289 |

Table 5: **Inference latency of different models.** Average forward-pass time on a RTX 4090 GPU.

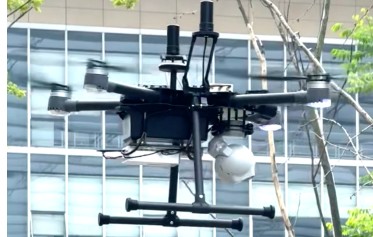

Figure 13: **UAV deployment platform in real-world operation.**

## E  UAV Hardware Setup

The UAV deployment platform is equipped with a complete set of hardware components, featuring a wheelbase of 600 mm and a payload capacity of approximately 3.7 kg. It integrates a LiDAR sensor and a gimbal camera for environmental perception. To achieve high-precision localization, the system includes an RTK module, a GPS module, and a dual-antenna GNSS setup. Onboard computation is powered by an NVIDIA Jetson Orin NX module, which supports the execution of lightweight models and flight control. RTK communication and the deployment of larger models are managed via a mobile ground station equipped with high-performance computing capabilities.

## F  Details of the Continuous Motion Scheme

**Globally-Aligned Continuous Motion Scheme.** As illustrated in Fig. 6, our model predicts future flight trajectories as relative displacements with respect to the drone's *local* coordinate system at the current time step. To guarantee spatial consistency and enable accurate long-horizon planning, we establish a *global* coordinate system anchored at the drone's initial position and orientation.

This global coordinate system provides a fixed spatial reference throughout the flight, allowing all predicted trajectories to be interpreted within a consistent coordinate frame.

At inference time, the drone's current pose is transformed into this global coordinate system, and the predicted relative displacements are accumulated accordingly. This design ensures that the planned trajectory remains precisely aligned with the drone's true motion history, maintaining global spatial coherence across time steps and mitigating drift or accumulated prediction errors.

**Look-Ahead Mechanism.** To further improve robustness during inference, we incorporate a *look-ahead mechanism*. As the drone moves forward, it may have already traversed certain predicted waypoints from earlier planning stages. To avoid redundant execution of these past waypoints, we employ a heuristic filtering strategy that dynamically discards the traversed ones, while retaining only the meaningful future targets. This mechanism enhances the smoothness and temporal continuity of the drone's motion, leading to more stable and efficient trajectory tracking in dynamic environments.

## G   Human Annotation and Flight Labor Cost

To support high-quality data collection and annotation, we hire experienced UAV pilots and professional annotators. Each pilot is responsible for operating UAVs in complex environments, executing flight instructions in Flow tasks. Annotators are tasked with reviewing, verifying, and correcting video-instruction pairs to ensure semantic alignment and accuracy. We pay both UAV pilots and annotation personnel $100 per hour. This rate is aligned with industry practices for skilled technical labor in robotics and machine learning data pipelines.

## H   Ethical Considerations

**Privacy Protection in Data Collection.** Prior to data collection, we obtained formal approval from campus administrators to ensure that the recording of buildings and environmental elements was fully authorized. During the collection process, we deliberately selected sparsely populated areas and avoided capturing bystanders or students to minimize the risk of inadvertent privacy exposure. In post-processing, all recorded images were carefully reviewed, and any potentially sensitive content—such as identifiable human faces, vehicle license plates, or signposts—was anonymized using standard blurring techniques. From the Supplementary Materials, all demo videos have undergone such blurring to safeguard personal privacy. The same anonymization procedure was applied consistently to all collected data to ensure responsible handling and compliance with privacy standards.

**Broader Impact and Misuse Prevention.** We acknowledge the potential risks associated with UAV technologies, including misuse for surveillance, military operations, or harassment. While our research focuses on advancing language-conditioned UAV control to improve intuitive human-drone interaction—primarily for applications such as aerial photography, inspection, and cinematography—the risks mentioned are general to UAV operation and not specific to our system. Nevertheless, to promote ethical alignment, we explicitly state that the released dataset and methods are intended solely for academic and non-commercial research. Appropriate license restrictions and usage guidelines will be included to prohibit any use for malicious or unethical purposes. We remain committed to responsible data stewardship and will continue to address community concerns related to data ethics and societal impact.

