# OpenReview forum: "UAV-Flow Colosseo: A Real-World Benchmark for Flying-on-a-Word UAV Imitation Learning"
_NeurIPS.cc/2025/Datasets_and_Benchmarks_Track — NeurIPS 2025 Datasets and Benchmarks Track poster_

### Official Review · Reviewer_ZHSV · 2025-06-28

**Rating:** 5
**Confidence:** 3

**Summary:**

This paper proposes a real-world benchmark, UAV-Flow, for language-conditioned, fine-grained UAV control. Authors collect flight trajectory data by employing human UAV pilots to operate UAV with a set of instructions in campus environment containing varied semantic elements. Then the collected data are aligned, manually filtered and LLM-based enriched to construct a diverse dataset. In the experiment authors investigate the performance of VLN and VLA models on the proposed benchmark.

**Additional Feedback:**

1. It seems that the baseline models are complex with large scales, they might need non-negligible time for inference. Can these models make timely control decisions within 0.2s (since the vedios are sampled at 5Hz)?

2. Would the UAV platform types and vedio qualities (e.g., 2K, 1080P or even lower) affect the performance of baseline models?

**Dataset Code Accessibility:**

Partly

**Dataset Code Comments:**

Authors provide the data of the proposed benchmark on huggingface, but I can not find the codes of baseline training or testing.

**Ethical Considerations:**

No, there are no or only very minor ethics concerns

**Final Justification:**

The authors address my concerns. This paper is a solid work, I think it should be accepted. So I keep my score.

**Limitations Weaknesses:**

1. The data in the benchmark is collected in ideal scenarios with bright and clear views, while in the real world the environment may not always be ideal. UAVs may be expected to perform atomic operations well at night or in rainy, windy, or foggy days, which could make UAV control challenging. Including complex real-world scenarios might enrich the benchmark and enhance the robustness of the trained models.

2. The data are collected using single UAV platform type, which might limit the data diversity and hinder the generalization of the trained models across different UAV platforms.

**Strengths Contributions:**

1. This paper proposes a fine-grained UAV control benchmark designed for imitation learning of atomic maneuvers guided by language instructions. It addresses the absence of benchmarks for low-level UAV control and advances the development of intelligent UAV systems.

2. The authors collect, align, filter and enrich a large-scale real-world dataset, which makes soild contributions to the community.

3. The paper is well-written and easy to follow.

---

> ### Author Rebuttal · Authors · 2025-07-30
>
> **Q1:** The data in the benchmark is collected in ideal scenarios with bright and clear views, while in the real world the environment may not always be ideal. UAVs may be expected to perform atomic operations well at night or in rainy, windy, or foggy days, which could make UAV control challenging. Including complex real-world scenarios might enrich the benchmark and enhance the robustness of the trained models.
>
> **A1:** We fully understand your concern. We have made a concerted effort to incorporate weather and lighting diversity during UAV data collection, including varying conditions such as clear, cloudy, overcast, and light rain, as well as different times of day ranging from early morning to noon and sunset.
>
> However, due to safety concerns, collecting data under extreme weather conditions is highly constrained. To ensure safe flight, we strictly follow both the UAV manufacturer's operational guidelines and local regulatory policies, which typically restrict or prohibit the operation of small UAVs under strong winds, heavy rain, or other hazardous conditions. Furthermore, flying in low-light or nighttime environments not only poses safety risks but also results in poor visual data quality, as the onboard RGB camera is limited in its ability to capture effective information under such conditions.
>
> Given these constraints, we have maximized the feasible range of weather and lighting diversity under safe and effective operating conditions. We hope that in future work, the availability of more advanced and robust sensor systems—capable of reliable perception in adverse weather and low-light settings—will enable further enhancement and expansion of environmental diversity in the dataset.
>
> **Q2:** The data are collected using single UAV platform type, which might limit the data diversity and hinder the generalization of the trained models across different UAV platforms.
>
> **A2:** Thank you for your suggestions. In fact, we have explicitly accounted for UAV platform diversity during data collection. To this end, we employed three different DJI camera drones: Mavic 3T, Air 3S, and Mini 4 Pro. These platforms vary in size, weight, sensor configurations, and flight capabilities, which substantially enhances the variability and representativeness of the collected dataset.
>
> In addition to data collection, we conducted real-world flight experiments to evaluate the cross-platform generalization of models trained on our dataset. Specifically, we tested the models on two additional UAV platforms: the DJI Tello and a custom-built UAV, with detailed specifications provided in Appendix E. The results show that our method generalizes well across heterogeneous UAV embodiments. We provide supporting visualizations and demonstration videos in supplementary materials.
>
> To further increase data diversity, we systematically varied the initial and target positions during trajectory collection and introduced a range of path geometries. This design strategy improves the model's adaptability and strengthens its generalization ability across diverse planning scenarios.
>
> **Q3:** Authors provide the data of the proposed benchmark on huggingface, but I can not find the codes of baseline training or testing.
>
> **A3:** We fully agree with your suggestion and are committed to open-sourcing the complete codebase of our models, along with the testing environment and real-world deployment pipeline.
>
> **Q4:** It seems that the baseline models are complex with large scales, they might need non-negligible time for inference. Can these models make timely control decisions within 0.2s (since the videos are sampled at 5Hz)?
>
> **A4:** We fully agree that inference latency is a crucial consideration for real-world UAV deployment. We have reported the inference time of all baseline models in Appendix D. Among them, the Pi-0-UAV model exhibits the longest inference time, approximately 0.289 seconds. While this slightly exceeds the typical 0.2-second control cycle, it is important to note that the Pi-0-UAV model generates an action chunk comprising a sequence of future actions that span the next 10 time steps, corresponding to 2 seconds of control trajectory. This design enables an inference-and-act closed-loop control scheme, which allows the system to operate within real-time constraints despite the longer per-inference latency.
>
> **Q5:** Would the UAV platform types and vedio qualities (e.g., 2K, 1080P or even lower) affect the performance of baseline models？
>
> **A5:** Thank you for your question. While the raw video data is recorded in 4K resolution, we adopt a standard preprocessing pipeline consistent with the visual encoders used in mainstream vision-language models such as CLIP and SigLIP, which typically operate on images resized to 224×224. All frames are therefore downsampled to a fixed resolution prior to training and inference, and our current experiments are conducted entirely on these downsampled inputs. Although higher-resolution inputs (e.g., 1080p, 2K, or 4K) may offer potential performance gains by preserving finer visual details, they also introduce substantial computational overhead. Our resolution choice reflects a practical balance between efficiency and accuracy, aligned with prevailing practices in large-scale vision-language model development.

---

> > ### Comment · Reviewer_ZHSV · 2025-08-03
> > **Thanks the authors**
> >
> > Thanks the authors for their follow-up, which addresses most of the concerns.
> >
> > [Q2, UAV platform concern] In the paper, I can only find Mavic 3T at line 111, the other platforms (Air 3S, Mini 4 Pro, DJI Tello and custom-built) are not mentioned. In Section 4.3 Real-World Deployment, authors does not introduce the UAV platform they use. Though Appendix E introduce the hardware setup of one UAV platform, the platform type is omitted. If authors use multiple UAV platforms, it should be clarified in the paper, otherwise readers may infer that experiments are conducted on a single UAV platform introduced in Appendix E.

---

> > > ### Author Response · Authors · 2025-08-03
> > > **Response to Reviewer ZHSV**
> > >
> > > Thank you very much for your constructive feedback!
> > >
> > > We indeed used three different UAV platforms during data collection and handled the variations in their log file structures accordingly. To avoid any confusion about the diversity of platforms involved, we will update the Real-World Data Collection section to explicitly list all UAV platforms used.
> > >
> > > For the real-world deployment experiments, we currently only include an image of the deployment setup in the supplementary materials. Your suggestion is very helpful— in the revised version, we will specify the deployment platform directly in the Real-World Deployment section and provide a more detailed description in the appendix by moving the Hardware Components for Deployment from the supplementary materials.
> > >
> > > Thank you again for your valuable suggestion!

---

> > > > ### Comment · Reviewer_ZHSV · 2025-08-08
> > > >
> > > > Thanks the authors for their responses, which address my concerns. This paper is a solid work, I think it should be accepted.

---

### Official Review · Reviewer_aHqt · 2025-07-02

**Ethics Flags:** Safety and security
**Rating:** 4
**Confidence:** 4

**Summary:**

This paper proposes a real-world benchmark (UAV-Flow) for language-guided fine-grained UAV control via imitation learning, with 30k+ trajectories and a ground-drone deployment framework. The "Flying-on-a-Word (Flow)" task paradigm is proposed, which shifts language interaction from traditional high-level path planning (VLN) to low-level fine trajectory control, filling the research gap of language-guided real-time response of UAVs.

**Dataset Code Accessibility:**

Partly

**Dataset Code Comments:**

Supplementary details are not very clear to follow.

**Ethical Comments:**

Purely machine based data, limited ethical issues. Only issue is whether the UAV data trained machine is safe enough.

**Ethical Considerations:**

No, there are no or only very minor ethics concerns

**Final Justification:**

Keep the original score as "Borderline accept"

**Limitations Weaknesses:**

1. The data diversity is a key issue from making real impact. Experiments and datasets are limited to campus environments. It's between to  at least have some diversity on validation set or have a separate section on simulation environments on generalized ability.
2. The experiment part is not very solid and supportive. I understand that methodology or a solid baseline is not the main contribution, but to validate the data quality and give better dataset understanding, it's still good to have a baseline result to give follow-up researchers more confidence on this data, especially considering this era.

**Strengths Contributions:**

1. The task formulation is novel: This paper proposes the Flying-on-a-Word (Flow) paradigm, shifting language-guided UAV control from high-level navigation (VLN) to fine-grained, reactive trajectory execution (e.g., "orbit a car" or "move at 45°"). It some how merge the traditional two-stage VL navigation and control into one unified task, and it could be a good fuel for VLA models.
2. Data construction seems solid: (30,692 trajectories across 5.02 km² of diverse campus environments). Combines RTK-GPS (cm accuracy), 4K FPV video, and synchronized 6-DoF states, could be a good real-world UAV data.
3. Complements with UAV-Flow-Sim, a UE-based simulation suite for controlled evaluation. Data diversity strategies (multi-startpoint executions, LLM-augmented language instructions) enhance generalization potential.

---

> ### Author Rebuttal · Authors · 2025-07-30
>
> **Q1:** The data diversity is a key issue from making real impact. Experiments and datasets are limited to campus environments. It's between to at least have some diversity on validation set or have a separate section on simulation environments on generalized ability.
>
> **A1:**
>
> 1. Data diversity
>
>     We fully considered data diversity during collection. The UAV-Flow dataset was collected across three campus environments (5.02 km²). A campus functions as a microcosm of society, approximating real-world urban scenarios with diverse elements like buildings, roads, parks. We enriched environmental diversity by including varied weather (clear, cloudy, overcast, light rain) and lighting (morning, noon, sunset) conditions. To ensure platform diversity, we used three DJI UAVs (Mavic 3T, Air 3S, Mini 4 Pro), and further broadened task diversity by varying start/target positions and path geometries.
>
> 2. Generalization testing
>
>     We sincerely appreciate your suggestion on generalization testing. In response, we selected two previously unseen scenarios—a village and an island—from the simulation environments, and constructed 100 test cases for each. These were used to evaluate the model’s generalization capability in unfamiliar settings, and the results are as follows:
>
>     |  Method  | Campus |          | Villege |          | Island |          |
>     | :------: | :----: | :------: | :-----: | :------: | :----: | :------: |
>     |          | **SR** | **NDTW** | **SR**  | **NDTW** | **SR** | **NDTW** |
>     | Seq2Seq-UAV  |  0.37  |  0.0001  |    0    |  0.0000  |   0    |  0.0000  |
>     |   CMA-UAV    |  4.76  |  0.0323  |    6    |  0.0176  |   5    |  0.0154  |
>     | Travel-UAV | 50.18  |  0.2507  |   39    |  0.2302  |   46   |  0.2474  |
>     | OpenVLA-UAV  | 61.17  |  0.3680  |   53    |  0.2723  |   55   |  0.2924  |
>     |   Pi-0-UAV   | 63.74  |  0.2535  |   55    |  0.1857  |   59   |  0.1887  |
>
>     The results show that when the model trained in the campus environment was transferred to the village and island settings, it still retained the ability to follow instructions, with only a slight drop in success rate (SR). One possible reason is that the campus environment contains some similar elements (e.g., buildings, roads, and vegetation), which helps the model partially adapt to new scenarios. However, a more noticeable decline in the NDTW metric was observed, indicating that the unfamiliar environments affected the model’s performance, leading to lower similarity between its trajectories and the ground-truth paths compared to known environments. We would be more than happy to include a detailed description of this evaluation in the paper.
>
> **Q2:** The experiment part is not very solid and supportive. I understand that methodology or a solid baseline is not the main contribution, but to validate the data quality and give better dataset understanding, it's still good to have a baseline result to give follow-up researchers more confidence on this data, especially considering this era.
>
> **A2:** We fully understand your concern. Although our baselines are built upon existing open-source VLN and VLA methods, we do more than merely adopt them: we carefully select representative frameworks, introduce substantial task-specific modifications, and conduct comprehensive training on the UAV-Flow dataset. Through systematic baseline selection, domain adaptation, and real-world deployment, we establish a meaningful and reliable evaluation framework tailored to UAV scenarios.
>
> 1. Baseline selection
>
>     In selecting the baselines, we emphasized both comprehensiveness and convincingness, covering representative frameworks for both navigation and manipulation grounded in vision–language models. This includes traditional VLN algorithms, as well as three types of VLM-based action prediction methods: the action-feature-token model (Travel)[1], the autoregressive action prediction model (OpenVLA)[2], and the flow-matching-based denoising model (Pi-0)[3]. We believe that this baseline system is both comprehensive and reliable, providing a solid foundation for evaluating UAV-oriented embodied intelligence.
>
> 2. Domain adaptation
>
>     To adapt existing methods to the UAV scenario, for VLN architectures, we modified the original discrete action classification mechanism into a trajectory point regression task. For VLA models, we redefined the action dimensions and adjusted the input structure and training objective functions to accommodate the perception and control requirements of the UAV-Flow task. These adaptations preserve the advantages of the original architecture while enabling effective alignment with the UAV-Flow task. Within our evaluation framework, the adapted methods reveal distinct performance characteristics and differences across approaches.
>
> 3. Real-world deployment
>
>     In addition, we have successfully deployed LLM-based models for real-world language-interactive UAV control, which we believe represents a meaningful innovation in the methodology part.
>
> We are committed to open-sourcing the complete codebase of our models, along with the testing environment and real-world deployment pipeline, providing the community with reliable benchmarks and practical tools to advance research on UAV-oriented embodied intelligence.
>
> **Q3:** Only issue is whether the UAV data trained machine is safe enough.
>
> **A3:** We appreciate your concern, as safety is indeed a critical issue in real-world UAV experiments. Throughout the data collection process, we ensured that all UAV flight data was collected in safe and controlled environments. The drone navigated without any collisions with surrounding obstacles.
>
> However, given the current capability limitations of the model, similar to other embodied AI systems, current data-driven models alone cannot guarantee absolute safety in real-world operations.
>
> To further enhance safety, we combined our data-driven models with rule-based or traditional obstacle avoidance controllers (e.g., ego-planner[4], which takes trajectory points as input), allowing the UAV to stop or autonomously avoid obstacles when encountering potential hazards. We believe that future improvements—through better model architecture design, the expansion of concer-case data, and deeper integration with traditional obstacle avoidance modules—will further enhance the safety and reliability of UAV systems.
>
> **Q4:** Supplementary details are not very clear to follow.
>
> **A4:** Thank you for your feedback. We will further refine the supplementary details to make them clearer and easier to follow. In addition, as mentioned in our response to Q2, we are committed to open-sourcing the complete codebase of our models, along with the testing environment and real-world deployment pipeline  to facilitate further research.
>
> [1] Towards realistic uav vision-language navigation: Platform, benchmark, and methodology
>
> [2] Openvla: An open-source vision-language-action model
>
> [3] π0: A Vision-Language-Action Flow Model for General Robot Control
>
> [4] EGO-Planner: An ESDF-Free Gradient-Based Local Planner for Quadrotors

---

### Official Review · Reviewer_rHFT · 2025-07-03

**Ethics Flags:** Data privacy, copyright, and consent
**Rating:** 5
**Confidence:** 3

**Summary:**

This paper introduces UAV-Flow Colosseo, a real-world benchmark for language-conditioned UAV imitation learning. In contrast to prior benchmarks that primarily focus on high-level planning and navigation of UAVs, the proposed benchmark aims to evaluate the models' ability for language-guided fine-grained trajectory control. The benchmark introduces an overall formulation of UAV imitation learning, a large-scale imitation learning dataset collected from three university campuses, and an evaluation suite. The authors benchmarked both traditional VLN and modern VLA models on the proposed Flow tasks, showing that VLA models are more suitable for fine-grained trajectory control of UAVs.

**Additional Feedback:**

Overall, at this point, I think this is a good submission that may inspire much future work in the intersection of UAV control and language-conditioned control. However, I think more implementation details could be elaborated by the authors.

**Dataset Code Accessibility:**

Yes

**Dataset Code Comments:**

The authors have open-sourced all data, simulation and testing environments, and model training procedures.

**Ethical Comments:**

The authors have open-sourced all data, simulation and testing environments, and model training procedures. Compensation details for drone pilots and instruction annotations are also reported, which look fine to me. The only potential risk I could think of is that since the training data is collected in university campuses, it may naturally contain personal information. The authors are encouraged to discuss the potential consequences and (if necessary) solutions for this point.

**Ethical Considerations:**

Yes, there are ethics concerns that require attention by the authors

**Final Justification:**

The authors have addressed most of my concerns in the rebuttal, and overall I think this is a good paper that should be accepted.

**Limitations Weaknesses:**

- In Section 3.1, the proposed "Globally-Aligned Continuous Motion scheme with a look-ahead mechanism" only has a cursory introduction.
- The proposed closed-loop simulation testing environment also lacks details. In general, I feel that the whole Section 3 could be significantly expanded to incorporate more details, while some fine details in Section 2 could be deferred to the appendix for better readability.

**Strengths Contributions:**

- The proposed UAV-Flow benchmark focuses on fine-grained, language-conditioned UAV control, which is a relevant problem and has been underexplored by prior work.
- Empirical evaluation is comprehensive, covering both traditional and modern UAV control paradigms.
- The paper is in general well-written and easy to follow, except that some implementation details need further elaboration (see "Limitations Weaknesses" for more details).

---

> ### Author Rebuttal · Authors · 2025-07-30
>
> **Q1:** In Section 3.1, the proposed "Globally-Aligned Continuous Motion scheme with a look-ahead mechanism" only has a cursory introduction.
>
> **A1:** We would be glad to provide a detailed explanation of the proposed Globally-Aligned Continuous Motion scheme with a look-ahead mechanism. Overall, this scheme establishes a unified spatial position for trajectory generation and dynamically adapts predictions as the drone progresses, ensuring globally consistent and robust flight planning.
>
> 1. Globally-Aligned Continuous Motion scheme.
>
>     As illustrated in Figure 6, the model predicts future flight trajectories as relative displacements with respect to the drone’s local coordinate system at the current time step. To ensure spatial consistency and enable accurate planning, we establish a global coordinate system anchored at the drone’s initial position and orientation. This global coordinate system provides a fixed reference throughout the flight, allowing all predicted trajectories to be interpreted in a consistent spatial context. At inference time, the drone’s current pose is transformed into this global coordinate system, and the predicted displacements are applied accordingly. This design ensures that the planned trajectory remains aligned with the drone’s true motion history, enabling precise and coherent execution across time steps.
>
> 2. Look-ahead mechanism.
>
>     To further enhance robustness during inference, a look-ahead mechanism is incorporated. As the drone moves forward, it may have already passed through certain predicted waypoints. To prevent redundant execution of these points, we employ a heuristic filtering strategy to dynamically discard traversed waypoints, retaining only meaningful future targets and improving the overall smoothness and reliability of trajectory following.
>
> **Q2:** The proposed closed-loop simulation testing environment also lacks details. In general, I feel that the whole Section 3 could be significantly expanded to incorporate more details, while some fine details in Section 2 could be deferred to the appendix for better readability.
>
> **A2:** Thank you for your valuable suggestions! We will carefully consider expanding both Section 3 and the Appendix to ensure sufficient detail.
>
> Our simulation environment is identical to the one used for collecting simulated data and is built upon the UnrealCV Zoo [1] for UAV simulation. We selected a campus-like environment featuring diverse elements such as buildings, trees, street lamps, cars, pedestrians, dogs, sculptures and fountains. A total of 273 test cases were constructed, with instruction types aligned with those presented in the experimental results shown in Figure 7.
>
> Each test case consists of a language instruction, a specified starting point, and a reference trajectory (used for computing the NDTW metric). During testing, each model undergoes closed-loop evaluation in the simulation environment, with a maximum inference step limit. The evaluation terminates when the model predicts stationary actions for several consecutive steps. Throughout the process, we record the complete inference trace. Finally, the performance of each model is assessed following the procedure described in Section 3.2.
>
> **Q3:** The only potential risk I could think of is that since the training data is collected in university campuses, it may naturally contain personal information. The authors are encouraged to discuss the potential consequences and (if necessary) solutions for this point.
>
> **A3:** We greatly appreciate your thoughtful concern regarding privacy.
>
> 1. Prior to data collection, we obtained formal approval from campus administrators, ensuring that the recording of buildings and environmental elements was fully authorized.
> 2. During data collection, we deliberately selected sparsely populated areas and avoided capturing bystanders or students, thereby minimizing the risk of inadvertent privacy exposure.
> 3. In post-processing, all images were carefully reviewed, and any potentially sensitive content—such as identifiable human faces—was anonymized via standard blurring techniques.
>
> We remain committed to responsible data stewardship and will promptly address any future concerns raised by the community.
>
> [1] UnrealZoo: Enriching Photo-realistic Virtual Worlds for Embodied AI

---

> > ### Comment · Reviewer_rHFT · 2025-08-02
> > **Reviewer's response**
> >
> > Thank you for your response, which addresses most of my concerns. I think incorporating these details into the revised manuscript would be helpful for future readers. Congratulations on the nice work!

---

> > > ### Author Response · Authors · 2025-08-02
> > > **Response to Reviewer rHFT**
> > >
> > > We sincerely appreciate your recognition and thoughtful feedback! Your comments are very helpful to us. We will incorporate your suggestions into the revised version and carefully polish the paper to further improve its quality.

---

### Note · Authors · 2025-08-13

We sincerely thank the AC and all reviewers for their constructive feedback, careful reading, and valuable suggestions.

First, we are very pleased that the reviewers recognize several strengths and contributions of our work, summarized as follows:

1. We propose the Flying-on-a-Word (Flow) paradigm, shifting language-guided UAV control from high-level navigation to fine-grained, reactive trajectory execution and filling the research gap of language-guided real-time control of UAVs.

2. We introduce a real-world dataset collected by professional UAV pilots, complemented by a simulation dataset for systematic evaluation under the Flow paradigm.

3. We design the UAV-Flow-Sim simulator for quantitative assessment, enabling fair and reproducible comparisons across different UAV control paradigms.

4. Many reviewers comment that our paper is well-structured, logically presented, and easy to follow.

We are also pleased that, during the rebuttal stage, we address many reviewer concerns in depth, including:

1. We provide additional details on data diversity and safety. The UAV-Flow dataset is collected across three campus environments under varied weather and lighting conditions. Three DJI camera drones are used to ensure platform diversity, and task diversity is achieved by varying start/target positions and path geometries. For legal compliance and privacy protection, we obtain formal authorization, collect data in sparsely populated areas, and anonymize any sensitive content through standard blurring techniques.

2. We supplement the description of simulation test environments and clarify our baseline selection criteria and domain adaptation strategies. Furthermore, we develop two entirely new simulation environments as part of an expanded evaluation framework.

3. We elaborate on our Globally-Aligned Continuous Motion scheme with a look-ahead mechanism. We demonstrate that real-time control is achievable even with large models, and we address reviewers’ concerns regarding deployment generalization across different UAV platforms and sensor configurations.

We sincerely appreciate the AC’s and reviewers’ recognition, constructive criticism, and insightful suggestions. We will incorporate all feedback into the revised version and carefully polish the manuscript. We hope that our complete Flow task setup—including dataset, baselines, simulation evaluation, and real-world deployment framework—contributes to advancing embodied intelligence for UAVs.

---

### Decision · Program_Chairs · 2025-09-18

**Decision:**

Accept (poster)

**Comment:**

The paper introduces the UAV-Flow benchmark for human-drone interaction, which all reviewers found to be an interesting and valuable contribution. I recommend accepting this paper.